# SNAS: STOCHASTIC NEURAL ARCHITECTURE SEARCH

**Sirui Xie, Hehui Zheng, Chunxiao Liu, Liang Lin**
SenseTime
{xiesirui, zhenghehui, liuchunxiao}@sensetime.com
linliang@ieee.org

## ABSTRACT

We propose *Stochastic Neural Architecture Search* (SNAS), an economical *end-to-end* solution to Neural Architecture Search (NAS) that trains neural operation parameters and architecture distribution parameters in same round of back-propagation, while maintaining the completeness and differentiability of the NAS pipeline. In this work, NAS is reformulated as an optimization problem on parameters of a joint distribution for the search space in a cell. To leverage the gradient information in generic differentiable loss for architecture search, a novel search gradient is proposed. We prove that this search gradient optimizes the same objective as reinforcement-learning-based NAS, but assigns credits to structural decisions more efficiently. This credit assignment is further augmented with locally decomposable reward to enforce a resource-efficient constraint. In experiments on CIFAR-10, SNAS takes fewer epochs to find a cell architecture with state-of-the-art accuracy than non-differentiable evolution-based and reinforcement-learning-based NAS, which is also transferable to ImageNet. It is also shown that child networks of SNAS can maintain the validation accuracy in searching, with which attention-based NAS requires parameter retraining to compete, exhibiting potentials to stride towards efficient NAS on big datasets.

## 1 INTRODUCTION

The trend to seek for state-of-the-art neural network architecture automatically has been growing since Zoph & Le (2016), given the enormous effort needed in scientific research. Normally, a Neural Architecture Search (NAS) pipeline comprises architecture sampling, parameter learning, architecture validation, credit assignment and search direction update.

There are basically three existing frameworks for neural architecture search. Evolution-based NAS like NEAT (Stanley & Miikkulainen, 2002) employs evolution algorithm to simultaneously optimize topology alongside with parameters. However, it takes enormous computational power and could not leverage the efficient gradient back-propagation in deep learning. To achieve the state-of-the-art performance as human-designed architectures, Real et al. (2018) takes 3150 GPU days for the whole evolution. Reinforcement-learning-based NAS is *end-to-end* for gradient back-propagation, among which the most efficient one, ENAS (Pham et al., 2018) learns optimal parameters and architectures together just like NEAT. However, as NAS is modeled as a Markov Decision Process, credits are assigned to structural decisions with temporal-difference (TD) learning (Sutton et al., 1998), whose efficiency and interpretability suffer from delayed rewards (Arjona-Medina et al., 2018). To get rid of the architecture sampling process, DARTS (Liu et al., 2019) proposes deterministic attention on operations to analytically calculate expectation at each layer. After the convergence of the parent network, it removes operations with relatively weak attention. Due to the pervasive non-linearity in neural operations, it introduces untractable bias to the loss function. This bias causes inconsistency between the performance of derived child networks and converged parent networks, thus parameter retraining comes up as necessary. A more efficient, more interpretable and less biased framework is in desire, especially for future full-fledged NAS solutions on large datasets.

In this work, we propose a novel, efficient and highly automated framework, *Stochastic Neural Architecture Search* (SNAS), that trains neural operation parameters and architecture distribution parameters in same round of back propagation, while maintaining the completeness and differentiability of the NAS pipeline. One of the key motivations of SNAS is to replace the feedback

mechanism triggered by *constant rewards* in reinforcement-learning-based NAS with more efficient gradient feedback from *generic loss*. We reformulate NAS with a new stochastic modeling to bypass the MDP assumption in reinforcement learning. To combine architecture sampling with computational graph of arbitrary differentiable loss, the search space is represented with a set of one-hot random variables from a fully factorizable joint distribution, multiplied as a mask to select operations in the graph. Sampling from this search space is made differentiable by relaxing the architecture distribution with *concrete distribution* (Maddison et al., 2016). We name gradients *w.r.t* their parameters *search gradient*. From a global view, we prove that SNAS optimizes the same objective as reinforcement-learning-based NAS, except the training loss is used as reward. Zooming in, we provide a policy gradient equivalent of this *search gradient*, showing how gradients from the loss of each sample are used to assign credits to structural decisions. By interpreting this credit assignment as Taylor Decomposition (Montavon et al., 2017a), we prove SNAS's efficiency over reinforcement-learning-based NAS. Additionally, seeing that existing methods (Liu et al., 2019) manually design topology in child networks to avoid complex architecture, we propose a global resource constraint to automate it, augmenting the objective with feasiblity concerns. This global constraint could be linearly decomposed for structural decisions, hence the proof of SNAS's efficiency still applies.

In our experiments, SNAS shows strong performance compared with DARTS and all other existing NAS methods in terms of test error, model complexity and searching resources. Specifically, SNAS discovers novel convolutional cells achieving 2.85±0.02% test error on CIFAR-10 with only 2.8M parameters, which is better than 3.00±0.14%-3.3M from 1st-order DARTS and 2.89%-4.6M from ENAS. It is also on par with 2.76±0.09%-3.3M from 2nd-order DARTS with fewer parameters. With a more aggressive resource constraint, SNAS discovers even smaller model achieving 3.10±0.04% test error on CIFAR-10 with 2.3M parameters. During the architecture search process, SNAS obtains a validation accuracy of 88% compared to around 70% of ENAS in fewer epochs. When validating the derived child network on CIFAR-10 without finetuning, SNAS maintains the search validation accuracy, significantly outperforming 54.66% by DARTS. These results validate our theory that SNAS is less biased than DARTS. The discovered cell achieves 27.3% top-1 error when transferred to ImageNet (mobile setting), which is comparable to 26.9% by 2nd-order DARTS.

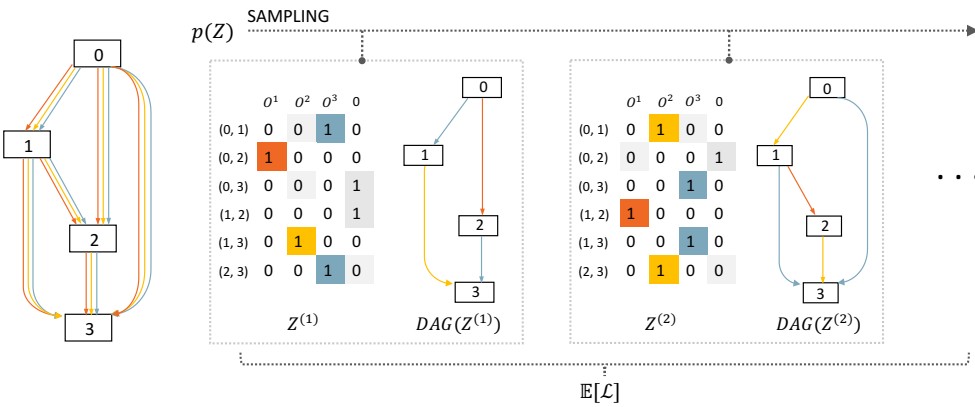

Figure 1: A conceptual visualization for a forward pass within SNAS. Sampled from $p(\boldsymbol{Z})$, $\boldsymbol{Z}$ is a matrix whose rows $\boldsymbol{Z}_{i,j}$ are one-hot random variable vectors indicating masks multiplied to edges $(i, j)$ in the DAG. Columns of this matrix correspond to operations $\boldsymbol{O}^k$. In this example, there are 4 operation candidates, among which the last one is *zero*, i.e. removing that edge. The objective is the expectation of generic loss $L$ of all *child graphs*.

## 2 METHODOLOGY

The main initiative of SNAS is to build an efficient and economical *end-to-end* learning system with as little compromise of the NAS pipeline as possible. In this section, we first describe how to sample from the search space for NAS in a cell, and how it motivates a stochastic reformuation for SNAS (Section 2.1). A new optimization objective is provided and the attention-based NAS's inconsistency is discussed. Then in Section 2.2, we introduce how this discrete search space is relaxed to be

continuous to let gradients back-propagate through. In Section 2.3, the search gradient of SNAS is connected to the policy gradient in reinforcement-learning-based NAS (Zoph & Le, 2016; Pham et al., 2018), interpreting SNAS's credit assignment with contribution analysis. At last, we introduce in Section 2.4 how SNAS automates the topology search to reduce the complexity of child netowrk, as well as how it decomposes this global constraint in the context of credit assignment.

## 2.1 SEARCH SPACE AND ARCHITECTURE SAMPLING

Searching for structure of a cell that is later stacked as building blocks for a deep architecture is an *ad hoc* solution to trade-off search efficiency and result optimality (Zoph et al., 2017; Liu et al., 2017a; Real et al., 2018; Pham et al., 2018; Liu et al., 2019). As shown in the left of Figure 1, the search space, *i.e.* a cell, is represented using a directed acyclic graph (DAG), which is called *parent graph*. Nodes $x_i$ in this DAG represent latent representation, whose dimensions are simply ignored to avoid abuse of notations. In convolutional networks, they are feature maps. Edges $(i, j)$ represent information flows and possible operations $\boldsymbol{O}_{i,j}$ to be selected between two nodes $x_i$ and $x_j$. To make the skip operation included, nodes are enforced to be ordered, while edges only point from lower indexed nodes to higher ones. Thus we have intermediate nodes

$$x_j = \sum_{i<j} \tilde{\boldsymbol{O}}_{i,j}(x_i), \tag{1}$$

where $\tilde{\boldsymbol{O}}_{i,j}$ is the selected operation at edge $(i, j)$. Analog to ENAS, SNAS search for operations and topology of this cell at the same time. Rather than using two distributions, this is done by introducing a *zero* operation, as in DARTS. Same as ENAS and DARTS, each cell is designed to have two inputs from the output of previous cells. The output of a cell is the concatenate of intermediate nodes.

Thanks to the fact that the volume of structural decisions, which pick $\tilde{\boldsymbol{O}}_{i,j}$ for edge $(i, j)$, is generally tractable in a cell, we represent it with a distribution $p(\boldsymbol{Z})$. Multiplying each one-hot random variable $\boldsymbol{Z}_{i,j}$ to each edge $(i, j)$ in the DAG, we obtain a *child graph*, whose intermediate nodes are

$$x_j = \sum_{i<j} \tilde{\boldsymbol{O}}_{i,j}(x_i) = \sum_{i<j} \boldsymbol{Z}_{i,j}^T \boldsymbol{O}_{i,j}(x_i). \tag{2}$$

In terms of how to parameterize and factorize $p(\boldsymbol{Z})$, SNAS is built upon the observation that NAS is a task with *fully delayed rewards* in a *deterministic environment*. That is, the feedback signal is only ready after the whole episode is done and all state transition distributions are delta functions. Therefore, a Markov Decision Process assumption as in ENAS may not be necessary. In SNAS, we simply assume that $p(\boldsymbol{Z})$ is fully factorizable, whose factors are parameterized with $\boldsymbol{\alpha}$ and learnt along with operation parameters $\boldsymbol{\theta}$. In Appendix A we connect the probability of a trajectory in the MDP of ENAS and this joint probability $p(\boldsymbol{Z})$.

Following the setting in Zoph & Le (2016), the objective of SNAS is also

$$\mathbb{E}_{\boldsymbol{Z} \sim p_{\boldsymbol{\alpha}}(\boldsymbol{Z})}[R(\boldsymbol{Z})]. \tag{3}$$

While the difference is that rather than using a constant reward from validation accuracy, we use the training/testing loss directly as reward, $R(\boldsymbol{Z}) = L_{\boldsymbol{\theta}}(\boldsymbol{Z})$, such that the operation parameters and architecture parameters can be trained under one generic loss:

$$\mathbb{E}_{\boldsymbol{Z} \sim p_{\boldsymbol{\alpha}}(\boldsymbol{Z})}[R(\boldsymbol{Z})] = \mathbb{E}_{\boldsymbol{Z} \sim p_{\boldsymbol{\alpha}}(\boldsymbol{Z})}[L_{\boldsymbol{\theta}}(\boldsymbol{Z})]. \tag{4}$$

The whole process of obtaining a Monte Carlo estimate of this objective is shown in Figure 1. An intuitive interpretation of this objective is to *optimize the expected performance of architectures sampled with* $p(\boldsymbol{Z})$. This differentiates SNAS from attention-based NAS like DARTS, which avoids the sampling process by taking analytical expectation at each edge over all operations. In Appendix B we illustrate the inconsistency between DARTS's loss and this objective, explaining its necessity of parameter finetuning or even retraining after architecture derivation. Resembling ENAS, SNAS does not have this constraint. We introduce in next subsection how SNAS calculates gradients *w.r.t* $\boldsymbol{\theta}$ and $\boldsymbol{\alpha}$.

## 2.2 PARAMETER LEARNING FOR OPERATIONS AND ARCHITECTURES

Though the objective (4) could be optimized with black-box gradient descent method as in Ranganath et al. (2014), it would suffer from the high variance of likelihood ratio trick (Williams, 1992) and could not make use of the differentiable nature of $L_{\theta}(\boldsymbol{Z})$. Instead, we use *concrete distribution* (Maddison et al., 2016) here to relax the discrete architecture distribution to be continuous and differentiable with reparameterization trick:

$$
\begin{aligned}
\boldsymbol{Z}_{i,j}^k &= f_{\boldsymbol{\alpha}_{i,j}}(\boldsymbol{G}_{i,j}^k) \\
&= \frac{\exp((\log \boldsymbol{\alpha}_{i,j}^k + \boldsymbol{G}_{i,j}^k)/\lambda)}{\sum_{l=0}^n \exp((\log \boldsymbol{\alpha}_{i,j}^l + \boldsymbol{G}_{i,j}^l)/\lambda)},
\end{aligned}
\tag{5}
$$

where $\boldsymbol{Z}_{i,j}$ is the softened one-hot random variable for operation selection at edge $(i,j)$, $\boldsymbol{G}_{i,j}^k = -\log(-\log(\boldsymbol{U}_{i,j}^k))$ is the $k$th *Gumbel* random variable, $\boldsymbol{U}_{i,j}^k$ is a uniform random variable. $\boldsymbol{\alpha}_{i,j}$ is the architecture parameter, which could depend on predecessors $\boldsymbol{Z}_{h,i}$ if $p(\boldsymbol{Z}_{i,j})$ is a conditional probability. $\lambda$ is the temperature of the softmax, which is steadily annealed to be close to zero in SNAS. In Maddison et al. (2016), it is proved that $p(\lim_{\lambda \to 0} \boldsymbol{Z}_{i,j}^k = 1) = \boldsymbol{\alpha}_{i,j}^k/(\sum_{l=0}^n \boldsymbol{\alpha}_{i,j}^l)$, making this relaxation unbiased once converged.

The full derivation of $\nabla \mathbb{E}_{\boldsymbol{Z} \sim p_{\boldsymbol{\alpha}}(\boldsymbol{Z})}[L_{\theta}(\boldsymbol{Z})]$ is given in Appendix C. Here with the surrogate loss $\mathcal{L}$ for each sample, we provide its gradient *w.r.t* $x_j$, $\boldsymbol{\theta}_{i,j}^k$ and $\boldsymbol{\alpha}_{i,j}^k$:

$$
\begin{aligned}
\frac{\partial \mathcal{L}}{\partial x_j} &= \sum_{m>j} \frac{\partial \mathcal{L}}{\partial x_m} \boldsymbol{Z}_m^T \frac{\partial \boldsymbol{O}_m(x_j)}{\partial x_j}, \\
\frac{\partial \mathcal{L}}{\partial \boldsymbol{\theta}_{i,j}^k} &= \frac{\partial \mathcal{L}}{\partial x_j} \boldsymbol{Z}_{i,j}^k \frac{\partial \boldsymbol{O}_{i,j}(x_i)}{\partial \boldsymbol{\theta}_{i,j}^k}, \\
\frac{\partial \mathcal{L}}{\partial \boldsymbol{\alpha}_{i,j}^k} &= \frac{\partial \mathcal{L}}{\partial x_j} \boldsymbol{O}_{i,j}^T(x_i)(\boldsymbol{\delta}(k'-k) - \boldsymbol{Z}_{i,j}) \boldsymbol{Z}_{i,j}^k \frac{1}{\lambda \boldsymbol{\alpha}_{i,j}^k}.
\end{aligned}
\tag{6}
$$

We name $\frac{\partial \mathcal{L}}{\partial \boldsymbol{\alpha}}$ *search gradient* similar to the one in Wierstra et al. (2008), even though no policy gradient is involved. This renders SNAS a differentiable version of evolutionary-strategy-based NAS.

## 2.3 CREDIT ASSIGNMENT

With the equivalence of $p(\boldsymbol{Z})$ in SNAS and $p(\tau)$ in ENAS from Section 2.1 and the *search gradient* of SNAS from Section 2.2, we discuss in this subsection what credits SNAS *search gradients* assign to each structural decision.

To assign credits to actions both temporally and laterally is an important topic in reinforcement learning (Precup, 2000; Schulman et al., 2015; Tucker et al., 2018; Xu et al., 2018). In ENAS, proximal policy optimization (PPO) (Schulman et al., 2017) is used to optimize the architecture policy, which distributes credits with TD learning and generalized advantage estimator (GAE) (Schulman et al., 2015). However, as the reward of NAS task is only obtainable after the architecture is finalized and the network is tested for accuracy, it is a task with *delayed rewards*. As proved by Arjona-Medina et al. (2018), *TD learning has bias with reward delay and corrects it exponentially slowly*.

Different from ENAS, there is no MDP assumption in SNAS, but the reward function is made differentiable in terms of structural decisions. From Section 2.2 we can derive the expected *search gradient* for architecture parameters at edge $(i,j)$:

$$
\mathbb{E}_{\boldsymbol{Z} \sim p(\boldsymbol{Z})}\left[\frac{\partial \mathcal{L}}{\partial \boldsymbol{\alpha}_{i,j}^k}\right] = \mathbb{E}_{\boldsymbol{Z} \sim p(\boldsymbol{Z})}\left[\nabla_{\boldsymbol{\alpha}_{i,j}^k} \log p(\boldsymbol{Z}_{i,j})[\frac{\partial \mathcal{L}}{\partial x_j} \tilde{\boldsymbol{O}}_{i,j}(x_i)]_c\right],
\tag{7}
$$

where $[\cdot]_c$ emphasizes $\cdot$ is constant for the gradient calculation *w.r.t.* $\boldsymbol{\alpha}$. A full derivation is provided in Appendix D. Apparently, the search gradient is equivalent to a policy gradient for distribution at this edge whose credit is assigned as

$$
R_{i,j} = -[\frac{\partial \mathcal{L}}{\partial x_j} \tilde{\boldsymbol{O}}_{i,j}(x_i)]_c.
\tag{8}
$$

From a decision-wise perspective, this reward could be interpreted as contribution analysis of $\mathcal{L}$ with Taylor Decomposition (Montavon et al., 2017a), which distributes importance scores among nodes in the same effective layer. Given the presence of skip connections, nodes may be involved into multiple effective layers, credits from which would be integrated. This integrated credit of a node $j$ is then distributed to edges $(i, j)$ pointing to it, weighted by $\tilde{O}_{i,j}(x_i)$. Details are given in Appendix E. Thus *for each structural decision, no delayed reward exists, the credits assigned to it are valid from the beginning*. This proves why SNAS is more efficient than ENAS. Laterally at each edge, credits are distributed among possible operations, adjusted with random variables $\mathbf{Z}_{i,j}$. At the beginning of the training, $\mathbf{Z}_{i,j}$ is continuous and operations share the credit, the training is mainly on neural operation parameters. With the temperature goes down and $\mathbf{Z}_{i,j}$ becomes closer to one-hot, credits are given to the chosen operations, adjusting their probabilities to be sampled.

### 2.4 Resource Constraint

Apart from training efficiency and validation accuracy, forwarding time of the child network is another concern in NAS in order for its feasible employment. In SNAS, this could be taken into account as a regularizer in the objective:

$$\mathbb{E}_{\mathbf{Z} \sim p_{\alpha}(\mathbf{Z})}[L_{\theta}(\mathbf{Z}) + \eta C(\mathbf{Z})] = \mathbb{E}_{\mathbf{Z} \sim p_{\alpha}(\mathbf{Z})}[L_{\theta}(\mathbf{Z})] + \eta \mathbb{E}_{\mathbf{Z} \sim p_{\alpha}(\mathbf{Z})}[C(\mathbf{Z})], \tag{9}$$

where $C(\mathbf{Z})$ is the cost of time for the child network associated with random variables $\mathbf{Z}$. Rather than directly estimating the forwarding time, there are three candidates from the literature (Gordon et al., 2018; Ma et al., 2018) that can be used to approximately represent it: 1) the parameter size ; 2) the number of float-point operations (FLOPs); and 3) the memory access cost (MAC). Details about $C(\mathbf{Z})$ in SNAS could be found in Appendix F.

However, not like $L_{\theta}(\mathbf{Z})$, $C(\mathbf{Z})$ is not differentiable *w.r.t.* either $\theta$ or $\alpha$. A natural problem to ask is, whether efficient credit assignment from $C(\mathbf{Z})$ could be done with similar decomposition introduced above, such that the proof of SNAS's efficiency still applies. And the answer is positive, thanks to the fact that $C(\mathbf{Z})$ is linear in terms of all one-hot random variables $\mathbf{Z}_{i,j}$:

$$C(\mathbf{Z}) = \sum_{i,j} C(\mathbf{Z}_{i,j}) = \sum_{i,j} \mathbf{Z}_{i,j}^T C(\mathbf{O}_{i,j}), \tag{10}$$

mainly because the size of feature maps at each node is not dependent on the structural decision. That is, the distribution at each edge $(i, j)$ is optimized with local penalty, which is the conservative decomposition of the global cost, consistent with the credit assignment principle in SNAS.

In SNAS, $p_{\alpha}(\mathbf{Z})$ is fully factorizable, making it possible to calculate $\mathbb{E}_{\mathbf{Z} \sim p_{\alpha}}[C(\mathbf{Z})]$ analytically with sum-product algorithm (Kschischang et al., 2001). Unfortunately, this expectation is non-trivial to calculate, we optimize the Monte Carlo estimate of the final form from sum-product algorithm

$$\mathbb{E}_{\mathbf{Z} \sim p_{\alpha}}[C(\mathbf{Z})] = \sum_{i,j} \mathbb{E}_{\mathbf{Z}_{\setminus i,j} \sim p_{\alpha}}[\mathbb{E}_{\mathbf{Z}_{i,j} \sim p_{\alpha}}[\mathbf{Z}_{i,j}^T C(\mathbf{O}_{i,j})]] \tag{11}$$

with policy gradients.

## 3 Experiments

Following the pipeline in DARTS, our experiments consist of three stages. First, SNAS is applied to search for convolutional cells in a small parent network on CIFAR-10 and we choose the best cells based on their search validation accuracy. Then, a larger network is constructed by stacking the learned cells (*child graphs*) and is retrained on CIFAR-10 to compare the performance of SNAS with other state-of-the-art methods. Finally, we show that the cells learned on CIFAR-10 is transferable to large datasets by evaluating their performance on ImageNet.

### 3.1 Architecture Search on CIFAR-10

**Motivation** We apply SNAS to find convolutional cells on CIFAR-10 for image classification. Unlike DARTS, which evaluates the performance of child networks during the searching stage by training their snapshots from scratch, we directly take the search validation accuracy as the performance evaluation criterion. This evaluation method is valid in SNAS since the searching is unbiased from its objective, as introduced in Section 2.1.

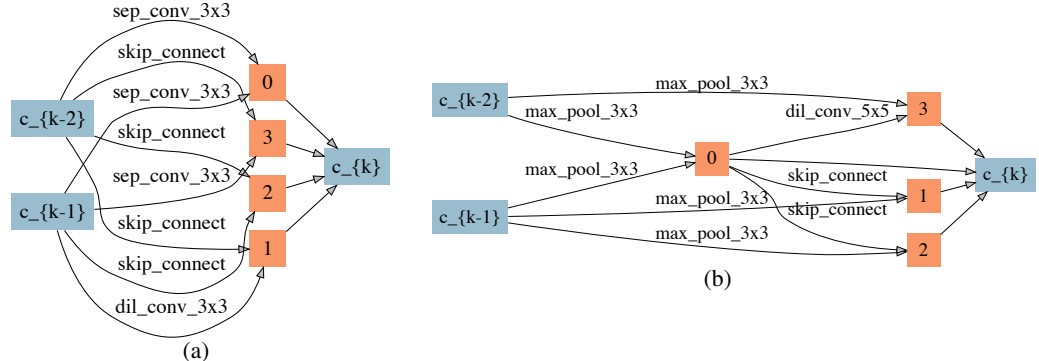

Figure 2: Cells (*child graphs*) SNAS (mild constraint) finds on CIFAR-10. (a) Normal cell. (b) Reduction cell.

**Dataset** CIFAR-10 dataset (Krizhevsky & Hinton, 2009) is a basic dataset for image classification, which consists of 50,000 training images and 10,000 testing images. Data transformation is achieved by the standard data pre-processing and augmentation techniques (see Appendix G.1).

**Search Space** Our setup follows DARTS, where convolutional cells (*parent graphs*) of 7 nodes are stacked for multiple times to form a network. The input nodes, *i.e.* the first and second nodes, of cell $k$ are set equal to the outputs of cell $k-2$ and cell $k-1$, respectively, with $1 \times 1$ convolutions inserted as necessary, and the output node is the depthwise concatenation of all the intermediate nodes. Reduction cells are located at the 1/3 and 2/3 of the total depth of the network to reduce the spatial resolution of feature maps. Therefore the architecture distribution parameters is $(\boldsymbol{\alpha}_{normal}, \boldsymbol{\alpha}_{reduce})$, where $\boldsymbol{\alpha}_{normal}$ is shared by all the normal cells and $\boldsymbol{\alpha}_{reduce}$ is shared by all the reduction cells. Details about all operations included are shown in Appendix G.1.

**Training Settings** In the searching stage, we train a small network stacked by 8 cells (*parent graphs*) using SNAS with three levels of resource constraint for 150 epochs. This network size is determined to fit into a single GPU. Single-level optimization is employed to optimize $\boldsymbol{\theta}$ and $\boldsymbol{\alpha}$ over the same dataset as opposed to bilevel optimization employed by DARTS. The rest of the setup follows DARTS (Appendix G.1). The search takes 32 hours[1] on a single GPU[2].

**Searching Process** The normal and reduction cells learned on CIFAR-10 using SNAS with mild resource constraint are shown in Figure 2. In Figure 3, we give the validation accuracy during the search of SNAS, DARTS and ENAS with 10 Randomly Generated Seeds. Comparing with ENAS, SNAS takes fewer epochs to converge to higher validation accuracy. Though DARTS converges faster than SNAS, this accuracy is inconsistent with the child network. Table 1 presents their comparison of the validation accuracy at the end of search and after architecture derivation without fine-tuning. While SNAS can maintain its performance, there is a huge gap between those two in DARTS.

Table 1: Search validation accuracy and child network validation accuracy of SNAS and DARTS. Results marked with * were obtained using the code publicly released by Liu et al. (2019).

| Architecture | Search Valid. Acc (%) | Child Net Valid. Acc (%) | Search Cost (GPU days) |
|---|---|---|---|
| DARTS (2nd order bi-level) (Liu et al., 2019)* | 87.67 | 54.66 | 1[3] |
| SNAS (single-level) + mild constraint | 88.54 | 90.67 | 1.5 |

---

[1]The batch size of SNAS is 64 and that of ENAS is 160.

[2]All the experiments were performed using NVIDIA TITAN Xp GPUs

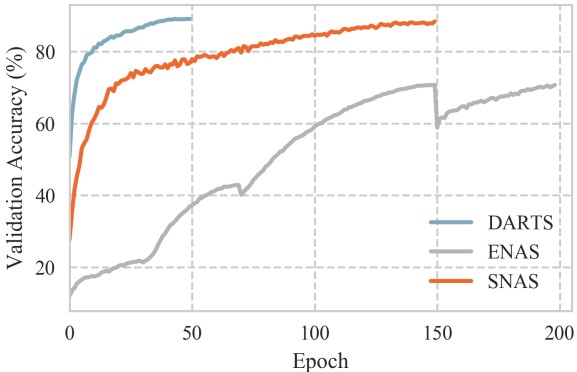 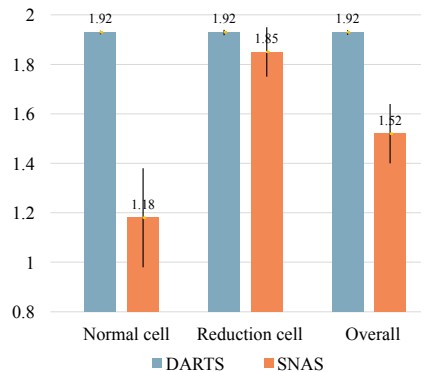

Figure 3: Search progress in validation accuracy from SNAS, DARTS and ENAS.

Figure 4: Entropy of architecture distribution in SNAS and DARTS.

This gap is caused by the extra architecture derivation step in DARTS, consisting of the following two steps. (1) Remove operations with relatively weak attention. As shown in Figure 4, the entropy of the architecture distribution (softmax) at each edge, *i.e.* $\mathcal{H}_{p_\alpha}$, is relatively high in DARTS, indicating its uncertainty in structural decisions. Hence removing other operations from the continuous relaxation will strongly affect the output of the network. (2) Remove relatively ambiguous edges. DARTS manually selects two inputs for each intermediate nodes, thus the topology is inconsistent with that in the training stage. While SNAS employs architecture sampling and resource regularizer to automatically induce sparsity. Phenomena shown in Figure 4 and Table 1 verify our claim that searching process in SNAS is less biased from the objective, *i.e.* Equation (4), and could possibly save computation resources for parameter retraining when extended to NAS on large datasets.

**Searching Results** Three levels of resource constraint, *mild, moderate* and *aggressive* are examined in SNAS. Mild resource constraint lies at the margin of the appearance of *zero* operation to drop edges in *child graphs*, as shown in Figure 2. Interestingly, every node takes only two input edges, just as in the designed scheme in ENAS and DARTS. When the constraint level is increased to moderate, the reduction cell begins to discover similar structures as normal cells, as shown in Appendix H. When a more aggressive resource constraint is added, the structure of reduction cells is further sparsified. As shown in Figure 5, more edges are dropped, leaving only two, which leads to the drop of some nodes, including the input node $c_{k-1}$, and two intermediate nodes $x_2$ and $x_3$. Note that this *child graph* is a structure that ENAS and DARTS are not able to discover [4].

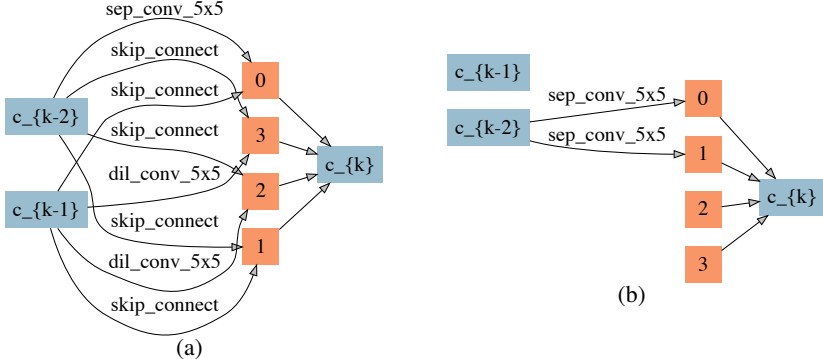

Figure 5: Cells (*child graphs*) SNAS (aggressive constraint) finds on CIFAR-10. (a) Normal cell. (b) Reduction cell.

---

[3]Repetition for convolutional cells is not necessary since the optimization outcomes are not initialization-sensetive (Liu et al., 2019).

[4]In the code from Liu et al. (2019), *zero* is omitted in *child graph* derivation as empirically it tends to learn the largest weight.

## 3.2 ARCHITECTURE EVALUATION ON CIFAR-10

**Motivation** In the searching stage, we follow the economical setup of DARTS to use only one single GPU, which constrains the parameter size of the child network. A conventional assumption in DARTS and ENAS[5] is that the final search validation accuracy has exploited the parameter size, the ceiling of which can only be raised by allowing more parameters. For a fair comparison, we follow this assumption in evaluation stage, stacking more cells (*child graphs*) to build a deeper network. This network is trained from scratch as in DARTS and ENAS to report the performance of the cells learned by SNAS on CIFAR-10.

**Evaluation Settings** A large network of 20 cells is trained from scratch for 600 epochs with batch size 96. Other hyperparameters remain the same as those for architecture search. Additional enhancements are listed in Appendix G.2. The training takes 1.5 days on a single GPU with our implementation in PyTorch.

Table 2: Classification errors of SNAS and state-of-the-art image classifiers on CIFAR-10.

| Architecture | Test Error (%) | Params (M) | Search Cost (GPU days) | Search Method | NAS Pipeline Completeness |
|---|---|---|---|---|---|
| DenseNet-BC (Huang et al., 2017) | 3.46 | 25.6 | - | manual | - |
| NASNet-A + cutout (Zoph et al., 2017) | 2.65 | 3.3 | 1800 | RL | complete |
| AmoebaNet-A + cutout (Real et al., 2018) | $3.34 \pm 0.06$ | 3.2 | 3150 | evolution | complete |
| AmoebaNet-B + cutout (Real et al., 2018) | $2.55 \pm 0.05$ | 2.8 | 3150 | evolution | complete |
| Hierarchical Evo (Liu et al., 2017b) | $3.75 \pm 0.12$ | 15.7 | 300 | evolution | complete |
| PNAS (Liu et al., 2017a) | $3.41 \pm 0.09$ | 3.2 | 225 | SMBO | complete |
| ENAS + cutout (Pham et al., 2018) | 2.89 | 4.6 | 0.5 | RL | complete |
| Random search baseline‡ + cutout (Liu et al., 2019) | $3.29 \pm 0.15$ | 3.2 | 1 | random | - |
| DARTS (1st order bi-level) + cutout (Liu et al., 2019) | $3.00 \pm 0.14$ | 3.3 | 0.4 | gradient-based | incomplete |
| DARTS (2nd order bi-level) + cutout (Liu et al., 2019) | $2.76 \pm 0.09$ | 3.3 | 1 | gradient-based | incomplete |
| DARTS (single-level) + cutout (Liu et al., 2019) | $3.56 \pm 0.10$ | 3.0 | 0.3 | gradient-based | incomplete |
| SNAS (single-level) + mild constraint + cutout | 2.98 | 2.9 | 1.5 | gradient-based | complete |
| SNAS (single-level) + moderate constraint + cutout | $2.85 \pm 0.02$ | 2.8 | 1.5 | gradient-based | complete |
| SNAS (single-level) + aggressive constraint + cutout | $3.10 \pm 0.04$ | 2.3 | 1.5 | gradient-based | complete |

**Results** The CIFAR-10 evaluation results are presented in Table 2. The test error of SNAS is on par with the state-of-the-art RL-based and evolution-based NAS while using three orders of magnitude less computation resources. Furthermore, with slightly longer wall-clock-time, SNAS outperforms 1st-order DARTS and ENAS by discovering convolutional cells with both a smaller error rate and fewer parameters. It also achieves a comparable error rate compared to 2nd-order DARTS but with fewer parameters. With a more aggressive resource constraint, SNAS can sparsify the architecture even further to distinguish from ENAS and DARTS with only a slight drop in performance, which is still on par with 1st-order DARTS. It is interesting to note that with same single-level optimization, SNAS significantly outperforms DARTS. Bilevel optimization could be regarded as a data-driven meta-learning method to resolve the bias proved above, whose bias from the exact meta-learning objective is still unjustified due to the ignorance of separate child network derivation scheme.

## 3.3 ARCHITECTURE TRANSFERABILITY EVALUATION ON IMAGENET

**Motivation** Since real world applications often involve much larger datasets than CIFAR-10, transferability is a crucial criterion to evaluate the potential of the learned cells (*child graphs*) (Zoph et al., 2017). To show whether the cells learned on by SNAS CIFAR-10 can be generalized to larger datasets, we apply the same cells evaluated in Section 3.2 to the classification task on ImageNet.

**Dataset** The *mobile* setting is adopted where the size of the input images is $224 \times 224$ and the number of multiply-add operations in the model is restricted to be less than 600M.

---

[5]As shown in the code publicly released by Pham et al. (2018)

Table 3: Classification errors of SNAS and state-of-the-art image classifiers on ImageNet.

| Architecture | Test Error (%) | | Params (M) | $+\times$ (M) | Search Cost (GPU days) | Search Method | NAS Pipeline Completeness |
|---|---|---|---|---|---|---|---|
| | top-1 | top-5 | | | | | |
| Inception-v1 (Szegedy et al., 2015) | 30.2 | 10.1 | 6.6 | 1448 | - | manual | - |
| MobileNet (Howard et al., 2017) | 29.4 | 10.5 | 4.2 | 569 | - | manual | - |
| ShuffleNet $2\times$ (v1) (Zhang et al.) | 26.4 | 10.2 | $\sim$5 | 524 | - | manual | - |
| ShuffleNet $2\times$ (v2) (Ma et al., 2018) | 25.1 | 10.1 | $\sim$5 | 591 | - | manual | - |
| NASNet-A (Zoph et al., 2017) | 26.0 | 8.4 | 5.3 | 564 | 1800 | RL | complete |
| NASNet-B (Zoph et al., 2017) | 27.2 | 8.7 | 5.3 | 488 | 1800 | RL | complete |
| NASNet-C (Zoph et al., 2017) | 27.5 | 9.0 | 4.9 | 558 | 1800 | RL | complete |
| AmoebaNet-A (Real et al., 2018) | 25.5 | 8.0 | 5.1 | 555 | 3150 | evolution | complete |
| AmoebaNet-B (Real et al., 2018) | 26.0 | 8.5 | 5.3 | 555 | 3150 | evolution | complete |
| AmoebaNet-C (Real et al., 2018) | 24.3 | 7.6 | 6.4 | 570 | 3150 | evolution | complete |
| PNAS (Liu et al., 2017a) | 25.8 | 8.1 | 5.1 | 588 | 225 | SMBO | complete |
| DARTS (Liu et al., 2019) | 26.9 | 9.0 | 4.9 | 595 | 1 | gradient-based | incomplete |
| SNAS (mild constraint) | 27.3 | 9.2 | 4.3 | 522 | 1.5 | gradient-based | complete |

**Evaluation Settings** We stack a network of 14 cells using the same cells designed by SNAS (mild constraint) and evaluated on CIFAR-10 (Section 3.2) and train it for 250 epochs with other hyperparameters following DARTS (see Appendix G.3). The training takes 12 days on a single GPU.

**Results** Table 3 presents the results of the evaluation on ImageNet and shows that the cell found by SNAS on CIFAR-10 can be successfully transferred to ImageNet. Notably, SNAS is able to achieve competitive test error with the state-of-the-art RL-based NAS using three orders of magnitude less computation resources. And with resource constraints added, SNAS can find smaller cell architectures that achieve competitive performance with DARTS.

## 4 RELATED WORKS

Improving the efficiency of NAS is a prerequisite to extending it to more complicated vision tasks like detection, as well as larger datasets. In the complete pipeline of NAS, parameter learning is a time-consuming one that attracts attention from the literature. Ideas to design auxiliary mechanisms like performance prediction (Baker et al., 2017; Deng et al., 2017), iterative search (Liu et al., 2017a), hypernetwork generated weights (Brock et al., 2017) successfully accelerate NAS to certain degrees. Getting rid of these auxiliary mechanisms, ENAS (Pham et al., 2018) is the state-of-the-art NAS framework, proposing parameter sharing among all possible *child graphs*, which is followed by SNAS. In Section 2 we introduced SNAS's relation with ENAS in details. Apart from ENAS, we are also inspired by Louizos et al. (2017) to use continuous distribution for structural decision at each edge and optimize it along with an $l_0$ complexity regularizer.

The most important motivation of SNAS is to leverage the gradient information in generic differentiable loss to update architecture distribution, which is shared by DARTS (Liu et al., 2019). In Section 2 and Appendix B we have introduced SNAS's advantage over DARTS, a reward for maintaining the completeness of the NAS pipeline. Actually, the idea to make use of this gradient information to improve the learning efficiency of a stochastic model has been discussed in the literature of generative model (Gu et al., 2015; Maddison et al., 2016) and reinforcement learning (Schmidhuber, 1990; Arjona-Medina et al., 2018). But as far as we known, we are the first one to combine the insights from these two fields to discuss possible efficiency improvement of NAS.

## 5 CONCLUSION

In this work, we presented SNAS, a novel and economical *end-to-end* neural architecture search framework. The key contribution of SNAS is that by making use of gradient information from generic differentiable loss without sacrificing the completeness of NAS pipeline, stochastic architecture search could be more efficient. This improvement is proved by comparing the credit assigned by the *search gradient* with reinforcement-learning-based NAS. Augmented by a complexity regu-

larizer, this *search gradient* trades off testing error and forwarding time. Experiments showed that SNAS searches well on CIFAR-10, whose result could be transferred to ImageNet as well. As a more efficient and less-biased framework, SNAS will serve as a possible candidate for full-fledged NAS on large datasets in the future.

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

## A  CONNECTING $p(Z)$ IN SNAS AND $p(\tau)$ IN ENAS

In ENAS, the NAS task is defined as an MDP, where the observation $o_i = a_0, a_1...a_{i-1}$. Thus the transition probability

$$p(o_i|o_{i-1}, a_{i-1}) = p(o_i|a_0, a_1...a_{i-2}, a_{i-1}) = \delta(a_0, a_1...a_{i-1}). \quad (12)$$

With the policy of RNN controller denoted as $\pi(a_i|o_i)$, the joint probability of a trajectory $\tau$ in this MDP is

$$
\begin{aligned}
p(\tau) &= \rho(o_0) \prod^i \pi(a_i|o_i) \prod^i p(o_{i+1}|o_i, a_i) \\
&= \prod^i \pi(a_i|o_i) \\
&= \prod^i \pi(a_i|a_0, a_1...a_{i-1}) \\
&= p(\boldsymbol{a}),
\end{aligned}
\quad (13)
$$

where $\boldsymbol{a}$ is a vector of all structural decisions, which is denoted as $\boldsymbol{Z}$ in SNAS. So we have

$$p(\tau) = p(\boldsymbol{Z}). \quad (14)$$

Note that if we factorize $p(\boldsymbol{Z})$ with conditional probability to have Markovian property as in Equation 13, we have the factor

$$p(\boldsymbol{Z}_i|\boldsymbol{Z}_0, \boldsymbol{Z}_1...\boldsymbol{Z}_{i-1}) = \pi(a_i|a_0, a_1...a_{i-1}). \quad (15)$$

## B  DIFFERENCE BETWEEN SNAS AND DARTS

We take a search space with three intermediate nodes for example to exhibit the difference between SNAS and DARTS (Liu et al., 2019), as shown in Figure 6. This search space could be viewed as a unit search space whose property could be generalized to larger space since it contains nodes in series and in parallel.

The objective of a NAS task is

$$\mathbb{E}_{\boldsymbol{Z} \sim p_{\boldsymbol{\alpha}}(\boldsymbol{Z})}[R(\boldsymbol{Z})], \quad (16)$$

where $p_{\boldsymbol{\alpha}}(\boldsymbol{Z})$ is the distribution of architectures, which is previously solved with reinforcement learning. In both SNAS and DARTS, the reward function is made differentiable using the training/testing loss, $R(\boldsymbol{Z}) = L_{\boldsymbol{\theta}}((Z))$, such that the architecture learning could leverage information in the gradients of this loss and conduct together with operation parameters training:

$$\mathbb{E}_{\boldsymbol{Z} \sim p_{\boldsymbol{\alpha}}(\boldsymbol{Z})}[R(\boldsymbol{Z})] = \mathbb{E}_{\boldsymbol{Z} \sim p_{\boldsymbol{\alpha}}(\boldsymbol{Z})}[L_{\boldsymbol{\theta}}(\boldsymbol{Z})]. \quad (17)$$

As introduced in Appendix A, SNAS solves (16) with a novel type of factorization, without relying on the MDP assumption. Though independent assumption between edges would restrict the probability distribution, there is no bias introduced.

However, to avoid the sampling process and gradient back-propagation through discrete random variables, DARTS takes analytical expectation at the input of each node over operations at incoming edges and optimizes a relaxed loss with deterministic gradients. Take the cell in Figure 6 as a base case, the objective before this relaxation is

$$
\begin{aligned}
&\mathbb{E}_{\boldsymbol{Z} \sim p_{\boldsymbol{\alpha}}(\boldsymbol{Z})}[L_{\boldsymbol{\theta}}(\boldsymbol{Z}_{j,l}^T \boldsymbol{O}_{j,l}(\boldsymbol{Z}_{i,j}^T \boldsymbol{O}_{i,j}(x_i)) + \boldsymbol{Z}_{j,m}^T \boldsymbol{O}_{j,m}(\boldsymbol{Z}_{i,j}^T \boldsymbol{O}_{i,j}(x_i)))] \\
&= \mathbb{E}_{\boldsymbol{Z} \sim p_{\boldsymbol{\alpha}}(\boldsymbol{Z})}[L_{\boldsymbol{\theta}}(\sum_{m>j} \boldsymbol{Z}_{j,m}^T \boldsymbol{O}_{j,m}(\boldsymbol{Z}_{i,j}^T \boldsymbol{O}_{i,j}(x_i))].
\end{aligned}
\quad (18)
$$

DARTS relaxed this objective to

$$L_{\boldsymbol{\theta}}(\sum_{m>j} \mathbb{E}_{p_{\boldsymbol{\alpha}_{j,m}}}[\boldsymbol{Z}_{j,m}^T \boldsymbol{O}_{j,m}(\mathbb{E}_{p_{\boldsymbol{\alpha}_{i,j}}}[\boldsymbol{Z}_{i,j}^T \boldsymbol{O}_{i,j}(x_i)])]). \quad (19)$$

Considering that $\boldsymbol{O}(x)$ are *ReLU-Conv-BN* stacks as in ENAS (Pham et al., 2018), which are non-linear, this transformation introduces unbounded bias. Though it will not be perceivable in training, where the complete graph is used for accuracy validation, consistent this loss, the derived graph is never validated during training. Hence the training is inconsistent with the true objective *maximizing the expected performance of derived architectures*. After an architecture derivation introduced in DARTS, the performance falls enormously and the parameters need to be retrained.

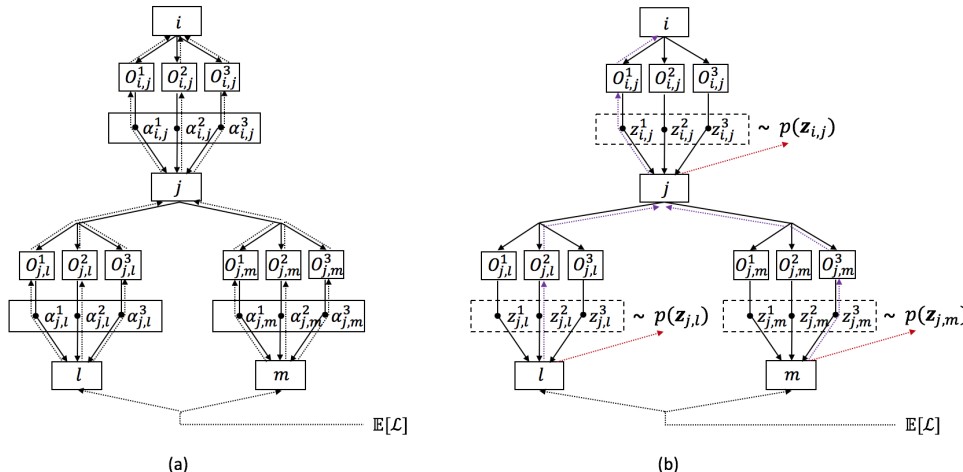

Figure 6: A comparison for gradients in DARTS and SNAS. (a) Deterministic gradients in DARTS; (b) Stochastic gradients in SNAS. Solid lines denote deterministic nodes, while dashed lines denote stochastic nodes. Black dotted lines denote compounded gradients, purple lines for parameter gradients in SNAS, red for *search gradients*.

## C   GRADIENTS IN SNAS

Figure 6(b) gives an illustration of a base three-intermediate-node unit in SNAS, where each edge has three operations (indexed by $k$) to choose from. In the search space of SNAS, intermediate nodes take input from all previous nodes. We have

$$x_j = \sum_{h<j} \boldsymbol{Z}_{h,j}^T \boldsymbol{O}_{h,j}(x_h) = \boldsymbol{Z}_{i,j}^T \boldsymbol{O}_{i,j}(x_i) + \sum_{h<i} \boldsymbol{Z}_{h,j}^T \boldsymbol{O}_{h,j}(x_h). \tag{20}$$

Let $\boldsymbol{\theta}_{i,j}^k$ be the parameters in $\boldsymbol{O}_{i,j}^k$, we have

$$\frac{\partial x_j}{\partial \boldsymbol{\theta}_{i,j}^k} = \boldsymbol{Z}_{i,j}^T \frac{\partial \boldsymbol{O}_{i,j}(x_i)}{\partial \boldsymbol{\theta}_{i,j}^k}. \tag{21}$$

As we use *concrete disctribution* here to make the sampling differentiable with reparametrization trick:

$$\begin{aligned} \boldsymbol{Z}_{i,j}^k &= f_{\boldsymbol{\alpha}_{i,j}}(\boldsymbol{G}_{i,j}^k) \\ &= \frac{\exp((\log \boldsymbol{\alpha}_{i,j}^k + \boldsymbol{G}_{i,j}^k)/\lambda)}{\sum_{l=0}^{n} \exp((\log \boldsymbol{\alpha}_{i,j}^l + \boldsymbol{G}_{i,j}^l)/\lambda)}, \end{aligned} \tag{22}$$

where $\boldsymbol{G}_{i,j}^k = -\log(-\log(\boldsymbol{U}_{i,j}^k))$ is the $k$th *Gumbel* random variable, $U_{i,j}^k$ is a uniform random variable, the gradient *w.r.t.* $\boldsymbol{\alpha}_{i,j}$ is:

$$\frac{\partial x_j}{\partial \boldsymbol{\alpha}_{i,j}^k} = \boldsymbol{O}_{i,j}^T(x_i) \frac{\partial f_{\boldsymbol{\alpha}_{i,j}}(\boldsymbol{G}_{i,j})}{\partial \boldsymbol{\alpha}_{i,j}^k}. \tag{23}$$

The partial derivative $\frac{\partial f_{\boldsymbol{\alpha}_{i,j}}}{\partial \boldsymbol{\alpha}_{i,j}^k}$ is

$$
\begin{aligned}
\frac{\partial f_{\boldsymbol{\alpha}_{i,j}}(\boldsymbol{G}_{i,j})}{\partial \boldsymbol{\alpha}_{i,j}^k} =& \frac{\frac{\partial}{\partial \boldsymbol{\alpha}_{i,j}^k} \exp((\log \boldsymbol{\alpha}_{i,j}^k + \boldsymbol{G}_{i,j}^k)/\lambda)}{\sum_{l=0}^n \exp((\log \boldsymbol{\alpha}_{i,j}^l + \boldsymbol{G}_{i,j}^l)/\lambda)}(\boldsymbol{\delta}(k'-k) - \frac{\exp((\log \boldsymbol{\alpha}_{i,j} + \boldsymbol{G}_{i,j})/\lambda)}{\sum_{l=0}^n \exp((\log \boldsymbol{\alpha}_{i,j}^i + \boldsymbol{G}_{i,j}^l)/\lambda)}) \\
=& \frac{\partial(\log \boldsymbol{\alpha}_{i,j}^k + \boldsymbol{G}_{i,j}^k)/\lambda}{\partial \boldsymbol{\alpha}_{i,j}^k} f_{\boldsymbol{\alpha}_{i,j}}(\boldsymbol{G}_{i,j}^k)(\boldsymbol{\delta}(k'-k) - f_{\boldsymbol{\alpha}_{i,j}}(\boldsymbol{G}_{i,j})) \\
=&(\boldsymbol{\delta}(k'-k) - f_{\boldsymbol{\alpha}_{i,j}}(\boldsymbol{G}_{i,j}))f_{\boldsymbol{\alpha}_{i,j}}(\boldsymbol{G}_{i,j}^k)\frac{1}{\lambda \boldsymbol{\alpha}_{i,j}^k} \\
=&(\boldsymbol{\delta}(k'-k) - \boldsymbol{Z}_{i,j})\boldsymbol{Z}_{i,j}^k \frac{1}{\lambda \boldsymbol{\alpha}_{i,j}^k}.
\end{aligned}
\tag{24}
$$

Substitute it back to (23), we obtain

$$
\frac{\partial x_j}{\partial \boldsymbol{\alpha}_{i,j}^k} = \boldsymbol{O}_{i,j}^T(x_i)(\boldsymbol{\delta}(k'-k) - \boldsymbol{Z}_{i,j})\boldsymbol{Z}_{i,j}^k \frac{1}{\lambda \boldsymbol{\alpha}_{i,j}^k}.
\tag{25}
$$

We can also derive $\frac{\partial x_m}{\partial x_j}$ for chain rule connection:

$$
\frac{\partial x_m}{\partial x_j} = \boldsymbol{Z}_{j,m}^T \frac{\partial \boldsymbol{O}_{j,m}(x_j)}{\partial x_j}.
\tag{26}
$$

Thus the gradient from the surrogate loss $\mathcal{L}$ to $x_j$, $\boldsymbol{\theta}_{i,j}^k$ and $\boldsymbol{\alpha}_{i,j}^k$ respectively are

$$
\begin{aligned}
\frac{\partial \mathcal{L}}{\partial x_j} &= \sum_{m>j} \frac{\partial \mathcal{L}}{\partial x_m} \boldsymbol{Z}_{j,m}^T \frac{\partial \boldsymbol{O}_{j,m}(x_j)}{\partial x_j}, \\
\frac{\partial \mathcal{L}}{\partial \boldsymbol{\theta}_{i,j}^k} &= \frac{\partial \mathcal{L}}{\partial x_j} \boldsymbol{Z}_{i,j}^k \frac{\partial \boldsymbol{O}_{i,j}(x_i)}{\partial \boldsymbol{\theta}_{i,j}^k}, \\
\frac{\partial \mathcal{L}}{\partial \boldsymbol{\alpha}_{i,j}^k} &= \frac{\partial \mathcal{L}}{\partial x_1} \boldsymbol{O}_{i,j}^T(x_i)(\boldsymbol{\delta}(k'-k) - \boldsymbol{Z}_{i,j})\boldsymbol{Z}_{i,j}^k \frac{1}{\lambda \boldsymbol{\alpha}_{i,j}^k}.
\end{aligned}
\tag{27}
$$

## D  CREDIT ASSIGNMENT FOR EQUIVALENT POLICY GRADIENT

From Appendix C we can see that the expected *search gradient* for architecture parameters at each edge is:

$$
\begin{aligned}
\mathbb{E}_{\boldsymbol{Z} \sim p(\boldsymbol{Z})}[\frac{\partial \mathcal{L}}{\partial \boldsymbol{\alpha}_{i,j}^k}] &= \mathbb{E}_{\boldsymbol{U} \sim Uniform}[\frac{\partial \mathcal{L}}{\partial x_j} \boldsymbol{O}_{i,j}^T(x_i) \frac{\partial f_{\boldsymbol{\alpha}_{i,j}}(-\log(-\log(\boldsymbol{U}_{i,j})))}{\partial \boldsymbol{\alpha}_{i,j}^k}] \\
&= \int_0^1 p(\boldsymbol{U}_{i,j}) \frac{\partial \mathcal{L}}{\partial x_j} \boldsymbol{O}_{i,j}^T(x_i) \frac{\partial f_{\boldsymbol{\alpha}_{i,j}}(-\log(-\log(\boldsymbol{U}_{i,j})))}{\partial \boldsymbol{\alpha}_{i,j}^k} d\boldsymbol{U}_{i,j} \\
&= \frac{\partial}{\partial \boldsymbol{\alpha}_1^k} \int_0^1 p(\boldsymbol{U}_{i,j})[\frac{\partial \mathcal{L}}{\partial x_j} \boldsymbol{O}_{i,j}^T(x_i)]_c f_{\boldsymbol{\alpha}_{i,j}}(-\log(-\log(\boldsymbol{U}_{i,j}))) d\boldsymbol{U}_{i,j} \\
&= \frac{\partial}{\partial \boldsymbol{\alpha}_{i,j}^k} \int p(\boldsymbol{Z}_{i,j})[\frac{\partial \mathcal{L}}{\partial x_j} \boldsymbol{O}_{i,j}^T(x_i)]_c \boldsymbol{Z}_{i,j} d\boldsymbol{Z}_{i,j} \\
&= \int p(\boldsymbol{Z}_{i,j}) \frac{\partial \log p(\boldsymbol{Z}_{i,j})}{\partial \boldsymbol{\alpha}_{i,j}^k}[\frac{\partial \mathcal{L}}{\partial x_j} \boldsymbol{O}_{i,j}^T(x_i)\boldsymbol{Z}_{i,j}]_c d\boldsymbol{Z}_{i,j} \\
&= \mathbb{E}_{\boldsymbol{Z} \sim p(\boldsymbol{Z})}[\nabla_{\boldsymbol{\alpha}_{i,j}^k} \log p(\boldsymbol{Z}_{i,j})[\frac{\partial \mathcal{L}}{\partial x_j} \boldsymbol{O}_{i,j}^T(x_i)\boldsymbol{Z}_{i,j}]_c] \\
&= \mathbb{E}_{\boldsymbol{Z} \sim p(\boldsymbol{Z})}[\nabla_{\boldsymbol{\alpha}_{i,j}^k} \log p(\boldsymbol{Z}_{i,j})[\frac{\partial \mathcal{L}}{\partial x_j} \tilde{\boldsymbol{O}}_{i,j}(x_i)]_c],
\end{aligned}
\tag{28}
$$

where $[\cdot]_c$ denotes $\cdot$ is a constant for the gradient calculation *w.r.t.* $\boldsymbol{\alpha}$. Note that in this derivation we stop the gradient from successor nodes, with an independence assumption enforced in back-propagation.

## E TAYLOR DECOMPOSITION FOR CONTRIBUTION ANALYSIS

With $d$ neurons (pixels) $x_i$ in the same layer of a deep neural network, whose output is $f(\boldsymbol{x})$, Montavon et al. (2017a) decomposes $f(\boldsymbol{x})$ as a sum of individual credits for $x_i$. This decomposition is obtained by the first-order Taylor expansion of the function at some root point $\tilde{\boldsymbol{x}}$ for which $f(\tilde{\boldsymbol{x}}) = 0$:

$$f(\boldsymbol{x}) = \sum_{i=1}^{d} R_i(\boldsymbol{x}) + O(\boldsymbol{x}\boldsymbol{x}^T), \tag{29}$$

where the individual credits

$$R_i(\boldsymbol{x}) = \frac{\partial f}{\partial \boldsymbol{x}_i}|_{\boldsymbol{x}=\tilde{\boldsymbol{x}}}(\boldsymbol{x}_i - \tilde{\boldsymbol{x}}_i) \tag{30}$$

are first-order terms and $O(\boldsymbol{x}\boldsymbol{x}^T)$ is for higher-order information. When ReLU is chosen as the activation function, $O(\boldsymbol{x}\boldsymbol{x}^T)$ can be omitted (Montavon et al., 2017b). Thus ones can always find a root point $\tilde{\boldsymbol{x}} = \lim_{\epsilon \to 0} \epsilon \boldsymbol{x}$ that incidentally lies on the same linear region as point $\boldsymbol{x}$, in which case the function can be written as

$$f(\boldsymbol{x}) = \sum_{i=1}^{d} R_i(\boldsymbol{x}) = \sum_{i=1}^{d} \frac{\partial f}{\partial \boldsymbol{x}_i}\boldsymbol{x}_i. \tag{31}$$

Noticing the similarity between (8) and (31), we try using Taylor Decomposition to interpret the credit assignment in SNAS. Given a sample $x_0$, ones can iterate all effective layers of the DAG and distribute credits from network output $f$ among nodes $x_j$ in each layer. In Figure 1 for example, $DAG(\boldsymbol{Z}^{(1)})$ has 2 effective layers, while $DAG(\boldsymbol{Z}^{(2)})$ has 3 effective layers. Given the presence of the skip connection, nodes may be involved into multiple layers and thus obtain integrated credits

$$\frac{\partial f}{\partial x_j} = \sum_{m>j} \frac{\partial f}{\partial x_m} \frac{\partial \tilde{\boldsymbol{O}}_m(x_j)}{\partial x_j}, \tag{32}$$

*e.g.* $x_1$ in DAG(2) integrates credits from $x_2$ and $x_3$. According to (1), multiple edges $(i,j)$ are pointing to $j$, which decompose (32) as:

$$\hat{R}_{i,j} = \frac{\partial f}{\partial x_j} \tilde{\boldsymbol{O}}_{i,j}(x_i). \tag{33}$$

Adjusting the weight of this sample with $\partial \mathcal{L}/\partial f$ and taking the optimization direction into account, we have

$$R_{i,j} = -\frac{\partial \mathcal{L}}{\partial x_j} \tilde{\boldsymbol{O}}_{i,j}(x_i) \tag{34}$$

## F CANDIDATES FOR LOCAL RESOURCE CONSTRAINTS

In the case of a convolutional layer, $H$, $W$ and $f$, $k$ correspond to the output spatial dimensions and the filter dimensions respectively and we use $I, O$ to denote the number of input and output channels. Since group convolution is also adopted in this paper to reduce the computational complexity, $g$ is the number of groups.

Thus, the parameter size and the number of float-point operations (FLOPs) of a single convolutional layer is

$$\text{parameter size} = \frac{fkIO}{g} \tag{35}$$

$$\text{FLOPs} = \frac{HWfkIO}{g} \tag{36}$$

By assuming the computing device has enough cache to store the feature maps and the parameters, we can simplify the memory access cost (MAC) to be the sum of the memory access for the input/output feature maps and kernel weights (Ma et al., 2018).

$$\text{MAC} = HW(I + O) + \frac{fkIO}{g} \tag{37}$$

In SNAS, because all the operations on a single edge share the same output spatial dimensions and the input/output channels, FLOPs of a convolutional operation is directly proportional to its parameter size. And although the memory access cost for the input/output feature maps $HW(I+O)$ does not depend on the parameter size, since both are positively correlated to the number of layers used in the operation, we may say there is a positive correlation between MAC and the parameter size. Thus, when only considering the convolution operations, solely using the parameter size as the resource constraint is sufficient. However, in SNAS, we also have the pooling operation and the skip connection, which are parameter free. The equations to calculate the resource criteria of a pooling operation or a skip connection are as follows.

FLOPs of pooling:

$$\text{FLOPs} = HWfkIO \tag{38}$$

FLOPs of skip connection:

$$\text{FLOPs} = 0 \tag{39}$$

MAC of pooling and skip connection:

$$\text{MAC} = HW(I+O) \tag{40}$$

We can see that MAC is the same for pooling and skip connection since they need to access the same input/output feature maps, therefore, to distinguish between pooling and skip connection, FLOPs need to be included in the resource constraint. Similarly, to distinguish between skip connection and none (free, no operation), MAC also need to be included.

In conclusion, to construct a resource constraint which fully distinguishes the four types of operations, all three locally decomposable criteria, the parameter size, FLOPs and MAC, need to be combined.

## G  DETAILED SETTINGS OF EXPERIMENTS

### G.1  ARCHITECTURE SEARCH ON CIFAR-10

**Data Pre-processing and Augmentation Techniques**  We employ the following techniques in our experiments: centrally padding the training images to $40 \times 40$ and then randomly cropping them back to $32 \times 32$; randomly flipping the training images horizontally; normalizing the training and validation images by subtracting the channel mean and dividing by the channel standard deviation.

**Implementation Details of Operations**  The operations include: $3 \times 3$ and $5 \times 5$ separable convolutions, $3 \times 3$ and $5 \times 5$ dilated separable convolutions, $3 \times 3$ max pooling, $3 \times 3$ average pooling, skip connection and *zero* operation. All operations are of stride one (excluded the ones adjacent to the input nodes in the reduction cell, which are of stride two) and the convolved feature maps are padded to preserve their spatial resolution. Convolutions are applied in the order of ReLU-Conv-BN, and the depthwise separable convolution is always applied twice (Zoph et al., 2017; Real et al., 2018; Liu et al., 2017a; 2019).

**Detailed Training Settings**  We follow the training settings as in Liu et al. (2019). The neural operation parameters $\boldsymbol{\theta}$ are optimized using momentum SGD, with initial learning rate $\eta_{\boldsymbol{\theta}} = 0.025$ (annealed down to zero following a cosine schedule), momentum 0.9, and weight decay $3 \times 10^{-4}$. The architecture distribution parameters $\boldsymbol{\alpha}$ are optimized by Adam, with initial learning rate $\eta_{\boldsymbol{\alpha}} = 3 \times 10^{-4}$, momentum $\beta = (0.5, 0.999)$ and weight decay $10^{-3}$. The batch size employed is 64 and the initial number of channels is 16.

### G.2  ARCHITECTURE EVALUATION ON CIFAR-10

**Additional Enhancement Techniques**  Following existing works (Zoph et al., 2017; Liu et al., 2017a; Pham et al., 2018; Real et al., 2018; Liu et al., 2019), we employ the following additional enhancements: cutout (DeVries & Taylor, 2017), path dropout of probability 0.2 (same as DARTS in the code publicly released by its authors) and auxiliary towers with weight 0.4.

### G.3 ARCHITECTURE TRANSFERABILITY EVALUATION ON CIFAR-10

**Detailed Training Settings** The network is trained with batch size 128, weight decay $3 \times 10^{-5}$ and initial SGD learning rate 0.1, which is decayed by a factor of 0.97 after each epoch. Auxiliary towers with weight 0.4 are adopted as additional enhancements.

## H CELLS LEARNED BY SNAS WITH A MODERATE RESOURCE CONSTRAINT

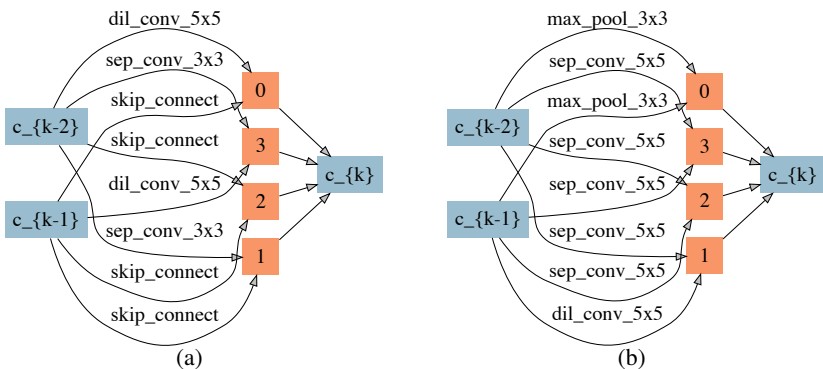

Figure 7: Cells (*child graphs*) SNAS (moderate constraint) finds on CIFAR-10. (a) Normal cell. (b) Reduction cell.

