# OpenReview forum: "SNAS: stochastic neural architecture search"
_ICLR.cc/2019/Conference_

### Official Review · AnonReviewer3 · 2018-11-01
**An incremental work on NAS with good experiment results.**

**Rating:** 7
**Confidence:** 4

**Review:**

This work refines the NAS method for efficient neural architecture search. The paper brings new methods for gradient/reward updates and credit assignment.

pros:
1. An improvement on gradient calculation and reward back-propagation mechanism
2. Good experiment results and fair comparisons

cons:
1. Missing details on how to use the gradient information to generate child network structures. In eq.2, multiplying each one-hot random variable Zij to each edge (i, j) in the DAG can obtain a child graph whose intermediate nodes are xj. However, it is still unclear how to generate the child graph. More details on generating child network based on gradient information is expected.
2. In SNAS, P(z) is assumed fully factorizable. Factors are parameterized with alpha and learnt along with operation parameters theta. The factorization of p(Z) is based on the observation that NAS is a task with fully delayed rewards in a deterministic environment. That is, the feedback signal is only ready after the whole episode is done and all state transitions distributions are delta functions. In eq. 3, the authors use the training/testing loss directly as reward, while the previous method uses a constant reward from validation accuracy. It is unclear why using the training/testing loss can improve the results?

---

> ### Author Response · Authors · 2018-11-21
> **Response to review**
>
> Thank you for your review.
>
> 1) How to use gradient information to generate child network
> SNAS does not directly use gradient information to generate child network. Search gradient is naturally applied to update architecture parameters, which are parameters for concrete distribution. Then in the derivation step, operations with largest probability in the concrete distribution are selected.
>
> 2) Why using training/testing loss as reward can improve the results?
> This is introduced with details in Section 2.3, as well as Appendix D and E, which we believe is one of our contribution. As stated in your comment, we first prove that NAS is a task in deterministic environment with fully delayed reward. Then a proof from [1] is introduced that TD-learning suffers from delayed bias when delayed reward exists. It is proposed and proved in [1] that Taylor decomposition of reward could resolve delayed bias because no temporal difference setting exists anymore. We then prove that the delayed reward in NAS could be decomposed and assigned to all structural decisions with gradient back-propagation, when differentiable training/testing loss is used as reward. Therefore, leveraging the proof from [1], we prove that SNAS should converge faster than ENAS, which is verified by our experiment as stated in Section 3.1. Intuitively speaking, we spot the unnecessary temporal difference setting in NAS and solve it by using training/testing loss as reward.
>
> [1] Arjona-Medina et al., "Rudder: Return decomposition for delayed rewards", arXiv 2018

---

### Official Review · AnonReviewer2 · 2018-11-03

**Rating:** 7
**Confidence:** 4

**Review:**

Summary:
This paper proposes Stochastic Neural Architecture Search (SNAS), a method to automatically and efficiently search for neural architectures. It is built upon 2 existing works on these topics, namely ENAS (Pham et al 2018) and DARTS (Liu et al 2018).

SNAS provides nice theory and explanation of gradient computations, unites the strengths and avoid the weaknesses of ENAS and DARTS. There are many details in the paper, including the Appendix. The idea is as follows:
+------------+---------------------+-------------------------+
| Method | Differentiable | Directly Optimize |
|                |                           |    NAS reward       |
+------------+---------------------+-------------------------+
| ENAS     |      No                |        Yes                   |
| DARTS   |      Yes               |        No                    |
| SNAS     |      Yes               |        Yes                   |
+------------+---------------------+-------------------------+
SNAS inherits the idea of ENAS and DARTS by superpositioning all possible architectures into a Directed Acyclic Graph (DAG), effectively sharing the weights among all architectures. However, SNAS improves over ENAS and DARTS as follows (Section 2.2):

1. SNAS improves over ENAS in that it allows independent sampling at edges in the shared DAG, leading to a more tractable gradient at the edges of the DAG, which in turn allows more tractable Monte Carlo estimation of the gradients with respect to the architectural parameters.

2. While DARTS also has the property (1), DARTS implements this by computing the expected value at each node in the DAG, with respect to the joint distribution of the input edges and the operations. This makes DARTS not optimize the direct NAS objective. SNAS, due to their smart manipulation of architectural gradients using Gumbel variables, still optimizes the same objective with NAS and ENAS, but has a smoother gradients.

Experimental results in the paper show that SNAS finds architectures on CIFAR-10 that are comparable to those found by ENAS and DARTS, using a reasonable amount of computing resource. These architectures can also be transferred to learn competent models on ImageNet, like those of DARTS. Furthermore, experimental observations (Figure 3) are consistent with the theory above, that is:

1. The search process of SNAS is more stable than that of ENAS (as SNAS samples with a smaller variance).
2. Architectures found by SNAS perform better than those of DARTS, as SNAS searches directly for the NAS reward of the sampled models.

Strengths:
1. SNAS unites the strengths and avoids the weaknesses of ENAS and DARTS

2. SNAS provides a nice theory, which is verified through their experimental results.

Weaknesses:
I don’t really have any complaints about this paper. Some presentations of the paper might have been improved, e.g. the discussion on the ZERO operation in other comments should have been included.

---

> ### Author Response · Authors · 2018-11-20
> **Response to review**
>
> Thank you very much for your positive comments and detailed summary!
>
> We have included experiments to show how the effect of ZERO op differentiates SNAS from DARTS. Please kindly have a check.

---

### Official Review · AnonReviewer1 · 2018-11-04
**Novel approach that addresses some shortcomings of the previous NAS techniques.**

**Rating:** 6
**Confidence:** 4

**Review:**

This paper improves upon ENAS and DARTS by taking a differentiable approach to NAS and optimizing the objective across the distribution of child graphs. This technique allows for end-to-end architecture search while constraining resource usage and allowing parameter sharing by generating effective reusable child graphs.

SNAS employs Gumbel random variables which gives it better gradients and makes learning more robust compared to ENAS. The use of Gumbel variables also allow SNAS to directly optimize the NAS objective which is an advantage over DARTS.

The resource constraint regularization is interesting. Regularizing on the parameters that describe the architecture can help constrain resource usage during the forward pass.

The proposed method is novel but the main concern here is that there is no clear win over existing techniques in terms of performance. I can't see anywhere in the tables where you demonstrate a clear improvement over DARTS or ENAS.

Furthermore, in your child network evaluation with CIFAR-10, you mention that the comparison is without fine-tuning. Do you think this might be contributing to the performance gap in DARTS?

---

> ### Author Response · Authors · 2018-11-20
> **Response to review (1)**
>
> Thank you very much for your review and questions!
>
> 1) Clear win over DARTS:
> As a NAS task, it is believed that the performance of a framework is evaluated with i) the efficiency and automation in searching process and ii) the accuracy and complexity of searching result, i.e. child networks. In this metric, SNAS's advantage over DARTS is three-fold, due to less-biased searching objective and the resource constraint:
>
> A. Less computing resources for the whole searching pipeline
> Child network derived from SNAS without any fine-tuning could maintain the accuracy, thus a) during searching, the accuracy in SNAS could reflect the performance of child network. DARTS, on the contrast, has to retrain the network for 100 epochs as stated in the caption of figure 3 to track the actual searching progress; b) after searching, DARTS has to retrain the child network even if there is no extension on cell number of channel number. SNAS, on the contrast, does not have this requirement. A retraining is only needed when the child network is extended, which in our work is basically for fair comparison. All these retraining will take much longer time when NAS is directly applied to a large dataset.
>
> B. Automated sparse network generation
> Though DARTS takes ZERO op, which represents deleting the edge, into account in the searching process, it is omitted in child network operation selection as discovered by one of our reviewers in this comment [1]. (This discovery is very interesting, as this reviewer discovered that ZERO tends to be the op with largest weight in DARTS. That is, in DARTS the "soft-2nd-max" is chosen.) The approach to delete edge is manually designed as "to choose the top-k incoming edges for each node". In SNAS, to keep or delete an edge is automatically learnt. That is to say, the ZERO op is acting its supposed job to engender sparsity. In our updated version, experiment showed that with an aggressive resource constraint, SNAS discovers architecture whose reduction cell has only two edges and two nodes but comparable accuracy with 1st-order DARTS in CIFAR-10, posing a question for the validity or optimality of manually designed scheme in DARTS.
>
> C. Comparable accuracy with less resource in child networks
> In our updated version, we show that with a moderate resource constraint which plays the role of a regularizer, SNAS discovers architecture with slightly better accuracy and fewer parameters comparing to 1st-order DARTS, which is also comparable to 2nd-order DARTS. Note that in this paper we only show result of 1st-order SNAS due to limited time and extensive experiment required, though the 2nd-order extension is straight-forward [2]. As shown in DARTS, as well as [3], 2nd-order empirically brings better optimality, a fair comparison would be with 1st-order DARTS. Actually in 1st-order SNAS, an accuracy comparable with 1st-order DARTS could be achieved with 1/3 fewer parameters, when an aggressive constraint is applied.
>
> As for transferring to ImageNet, there is no theoretical justification in the literature to the best of our knowledge, we provide it mainly for a fair comparison with DARTS. Our next step is to try a direct search on ImageNet leveraging that SNAS does not need retraining on the searching result.
>
>
> 2) The effect of fine-tuning in evaluating child networks directly derived from DARTS
> In our empirical study, a fine-tuning of the derived child networks can improve its performance, but could not remedy the gap completely after 100 epochs. (100 epoch is the plateau of fine-tuning, and also a fair comparison with SNAS.) And there seems always to be a small gap (-1.0+/-0.7)% between the accuracy after fine-tuning and at the end of searching.
>
> More importantly, the 'gap' we want to discuss here is between the performance of a derived child network and the optimization objective in searching. As shown in Figure 3 in our paper, the optimization objective in searching is already converged to some optimum before this derivation, for both architecture parameters and operation parameters. Theoretically speaking, to use this parent network would be a justified result for the optimization problem. But with the absence of a guarantee that softmax weights will become discrete in the end, it would become an attention learning task, rather than NAS task. Though there exist methods like designing prior or extra learning objective to autonomously encourage one-hot-ness, a scheme is manually designed to delete a large portion of operations even though their weights are not 0. A natural question to ask is, why in this case the architecture parameter is still the optimal, even though the performance of child networks could be boosted with some fine-tuning.
>
>
>
> [1] https://openreview.net/forum?id=rylqooRqK7&noteId=rkeruyjYhX
> [2] Finn et al., "Model-Agnostic Meta-Learning for Fast Adaptation of Deep Networks", ICML 2017.
> [3] https://openreview.net/forum?id=rylqooRqK7&noteId=BJxaZ7Kojm

---

> > ### Author Response · Authors · 2018-11-20
> > **Response to review (2)**
> >
> > 1.2) Clear win over ENAS
> > Follow the metric defined above, SNAS's advantage over ENAS is three-fold, due to a better credit assignment mechanism and a resource constraint:
> > A. Less epochs to converge to higher accuracy in searching;
> > B. Automated sparse network generation;
> > C. Slightly better accuracy with 1/3 fewer parameters in child networks.

---

> > > ### Author Response · Authors · 2018-11-26
> > > **Response to review (3) - Clarification on 1st-order optimization**
> > >
> > > We are sorry that in our last response we mistook 1st-order DARTS as single-level DARTS since the latter one was not reported by authors. It is reported in the newest version of DARTS, which is also added to our updated version. The "1st-order SNAS" in our last response actually meant single-level SNAS, because the neural operation parameters and architecture distribution parameters are updated simultaneously. In DARTS's newest version, it is stated that single-level DARTS performs much worse than bi-level, either 1st-order or 2nd-order, which is thus much worse than SNAS. This supports our claim that SNAS is less biased.
> > >
> > > And we can also provide an interpretation for 2nd-order DARTS's comparable performance with SNAS. From our understanding, DARTS is using meta-learning to look for a resolution for the bias proved by us in a data-driven way. Though authors cited [1], it could not justify that is optimizing the exact objective, for basically two reasons. Firstly, the connection between sufficient condition provided in [1] and DARTS is not discussed. Secondly, even if an explicit connection could be provided, 2nd-order DARTS is still biased due to the ignorance of the separate derivation scheme in the meta-learning loss (i.e. bi-level loss), which is proved by our experiments and single-level DARTS's unsatisfying performance.
> > >
> > > We admit that the possibility of improvement with bi-level optimization exists even in less biased methods like SNAS. And the rationale is that some operations like skip connection affects the loss in next iteration more than the one in current iteration. When first proposed in [2], the skip connection is expected to help gradients' back-propagation. That is, skip connection plays a role of hyper-parameter for the gradient update process, the optimization of which prefers meta-learning. Unfortunately, we don't have enough time to run experiments to validate this rationale. It will be our next future work.
> > >
> > > [1] Franceschi et al., "Bilevel programming for hyperparameter optimization and meta-learning". ICML 2018.
> > > [2] He et al., "Deep Residual Learning for Image Recognition", CVPR 2016.

---

### Public Comment · (anonymous) · 2018-10-22
**Missing detail**

It is unclear how the authors obtain the child network for DARTS. As is mentioned by the paper, the architecture derivation step in DARTS consists of two steps. (1) Remove operations with relatively weak attention and the zero operation. (2) Remove relatively ambiguous edges. As we know, these removed operations and edges make up a very large percentage of the softmax score. After the removal, the scale of the value of the nodes (output feature maps) drops significantly. Do we need to re-scale the value of output feature maps for compensation?  I cannot reproduce the result of 54.66% in Table 1.

---

> ### Author Response · Authors · 2018-10-23
> **On child network derivation**
>
> Thank you for the question.
>
> The derivation method for DARTS is stated in Section 2.4 in their paper, details could also be found in the implementation publicly released by authors. In our paper we paraphrase it to give an explanation to the drop in accuracy after this derivation. As stated in your comments, "these removed operations and edges make up a very large percentage of the softmax score". Through this comparison we want to emphasize SNAS's consistency in child network derivation, because of explicitly taking it, i.e. sampling, into account in the searching loss.
>
> In this comparison, child networks are directly tested after the derivation for both SNAS and our replication of DARTS. No re-scaling or any other extra transformation is involved. But we do find that by using the unrolled option, DARTS's result (54.66%) falls less than the non-unrolled one (34.37%). SNAS, in contrast, shows slightly better result (90.67%) after derivation than at the end of searching (88.54%), probably as the latter one is a Monte Carlo estimate of expectation.
>
> We will add more details into the revised version, thanks again for this comment.

---

### Public Comment · (anonymous) · 2018-11-02
**On ZERO operation**

I have tried to run the released code of DARTS and found that in the code, the operation ZERO is omitted during the derivation process. I also checked the logit of ZERO operation learned by DARTS, and found that in the normal cell, it is the largest in most edges.

Is the same derivation code used by SNAS and thus omitting ZERO operation in the experiments? If not, could you please give an explanation to:
1) whether the logit of ZERO is the largest in most edges of the normal cell learned by SNAS?
2) If the result is different from DARTS, why is this difference?

---

> ### Author Response · Authors · 2018-11-02
> **ZERO operation is just as other operations if no complexity constraint is added**
>
> Thank you for your interesting discovery and the questions.
>
> 0) derivation code for SNAS is the same as DARTS?
> No. We have implemented our derivation method to replace the one provided by DARTS as SNAS uses fundamentally different one. But in our replication of DARTS, we ran the implementation publicly released without checking the code for derivation. We are trying to replicate DARTS's result again, taking your claim into consideration.
>
> 1) is the logit of ZERO the largest in most edges of the normal cell?
> As introduced in Section 2.4 in our paper, SNAS employs a complexity loss to encourage sparsity in child network. This is different from ENAS and DARTS which directly select two input edges for each node. With a relatively large hyperparameter for this complexity loss, the logit of ZERO dominates some of the edges, though the child network still have other non-ZERO edges to keep it connected. If no complexity loss is added, the child network tends to remain the complete topology, which is actually one of our motivations to introduce complexity loss.
>
> 2) reason for this discrepancy
> Assuming that your result is valid, it is explained by DARTS‘s authors to be the underdetermined contribution and rescaling effect of ZERO in the mixed op. We have the following two hypotheses for the discrepancy between SNAS and DARTS, which would be added to revised version if we can prove them with mathematical deduction:
> a. different from the softmax attention in DARTS, SNAS employs Gumbel-Softmax, whose mechanism involves Gumbel random variables. With equivalent logit and temperature, the random variable vector from Gumbel-Softmax is possibly more one-hot than the deterministic softmax attention. The more discrete network could amplify the incapability of ZERO for a smaller loss, thus the logit of it would not be boosted.
> b. the gradients back-propagated through deterministic softmax and Gumbel-Softmax are different. As provided in Section 2.3 and Appendix C in our paper, there is stochasticity (random variables Z) involved in the search gradients in SNAS. Given the special trait of ZERO that O(x)=0, it is possible to be a local optimum or a gradient blackhole for deterministic softmax, which is probably escaped by Gumbel-Softmax with the stochasticity.

---

### Public Comment · (anonymous) · 2018-11-08
**Derivation of child network for SOTA comparison**

Hi authors!
I am so impressed by your paper because it feels like this is a connection between NAS and DARTS.
However, I am curious about some details on CIFAR10 evaluation.
Within a cell, there are 7 nodes right as DARTS right?
Then, how is the number of initial channels set? The same number of initial channels 16 applies as search mode?
If so, it seems to me that the number of parameters reported seems so large.
Could you specify the number of initial channels plus another hyperparmeters to clarify the evaluation setting?

Anyway, thanks for the nice paper!

---

> ### Author Response · Authors · 2018-11-08
> **On initial channel number**
>
> Thank you for the question.
>
> For fair comparison with DARTS, we employed the same set of hyperparameters as specified in the code publicly released by its authors. I bet the number of initial channels in your question refers to the channel number used for the first cell in the network according to the code. During CIFAR10 evaluation, the number of initial channels is set to 36 rather than 16. This will lead to an increase in the number of parameters.
>
> We will add more hyperparameter details in the revised version. Hope this can answer your question and thanks again for the comment.

---

> > ### Public Comment · (anonymous) · 2018-11-08
> > **More clarification**
> >
> > Thank you for your quick and kind reply!
> > To my knowledge, for CIFAR10 evaluation, the number of layers was set to 20(reported in the paper), the number of nodes within a cell was set to 7(adopted from search) and the number of initial channels was set to 36(in the comment above).
> > Then, the number of parameters should be much larger than 3.3M I think... Thus, think if the number of initial channels was set to 36, the number of layers should have been set to 8 as search mode to match the number of parameters reported in SOTA comparison table... Is it wrong?

---

> > > ### Author Response · Authors · 2018-11-11
> > > **On parameter size**
> > >
> > > Thank you for your question.
> > >
> > > We have implemented a counting function to reproduce the result of DARTS and used it to count the parameter size of our network.
> > >
> > > Do you mind showing how you reached the claim that the size should be much larger?

---

> > > > ### Public Comment · (anonymous) · 2018-11-12
> > > > **20 Layers + 36 initial channels -> Too much parameters**
> > > >
> > > > Thanks for your reply!! As I run the code in github of DARTS paper, the parameter size was about 7M when a recurrent cell with 7 intermediate nodes is evaluated under the setting of 20 layers and 36 initial channels. Thus, I thought I should have reduced the number of nodes or the number of layers whatever...

---

> > > > > ### Author Response · Authors · 2018-11-20
> > > > > **Further on parameter size**
> > > > >
> > > > > Thank you for your explanation.
> > > > >
> > > > > The experiments of SNAS are conducted only on convolutional cells due to limited time. Sorry we might not be able to answer questions regarding the size of the recurrent cells.

---

### Public Comment · (anonymous) · 2018-11-19
**Some question about the experiments.**

Hi authors.
This is an interesting work, but I feel that some experimental comparisons are a little bit unfair as follows:

1. In Table 2, the authors report DARTS* has a test error of 3.15%, which is claimed as reproduced by the released code. I also run their codes and can obtain a similar performance with 2.86%. Reporting 3.15% might be misleading to others readers, and make other researchers wrongly refer the results of DARTS in the following papers.

2. DARTS has conducted results on both CNN and RNN. Why SNAS does not report the RNN results?

3. Would you mind to give more explanation about Figure 4?

4. In Figure 3, how did you run ENAS? In addition, do these three methods use the same training and validation set? Based on the GitHub issue of DARTS, the validation accuracy of DARTS does not have an explicit connection with the performance of the discovered model. How about the connection between the validation accuracy of SNAS and the final discovered model of SNAS?

5. For NAS approaches, there is usually a variance of the performance of the final discovered model. Would you mind to report the results of three runs of SNAS?

---

> ### Author Response · Authors · 2018-11-20
> **Response to questions**
>
> Thank you for your questions.
>
> 1) Reported DARTS accuracy lower than original paper
> The purpose of providing this reproduced result is to evaluate the searching result from last subsection (Section 3.1). Sorry for the confusion it caused. We have removed it in the updated version. In our search we only achieve (3.02+/-0.14)% evaluation accuracy, which is still a bit lower than accuracy reported in original paper. We are sure that we ran experiments with correct hyper-parameters. But as mentioned in your Q5, in stochastic searching task, sometimes ones just need a bit of luck due to the stochastic nature of the objective. We are willing to reveal random seeds at request.
>
> 2) Empirical study on rnn
> We didn't have enough time to run extensive experiment on rnn. But we believe the theory proposed in our work could be sufficiently verified with our extensive experiments on cnn, whose extension to rnn would be straightforward.
>
> 3) An explanation on Figure 4.
> In the updated version, Figure 4 is updated to show stats of entropy of softmax weights at all edges in the searching result. With a lower entropy in the learnt parent graphs, SNAS is more certain on the structural decision.
>
> 4.1) Training setting
> Experiments for ENAS were run with the default setting. And we noticed that the parent network in DARTS is a little bit different from ENAS, though parameter size is quite close. The reported result for SNAS were run with setting of DARTS for fair comparison, given that for some reason DARTS could not be fit into one GPU card with ENAS's setting. But we have also run SNAS experiment with ENAS's setting, whose searching curve has only negligible difference from the reported one. All experiments were run on the same training and testing set.
>
> 4.2) Correlation of training accuracy and child network final accuracy
> For SNAS, the correlation coefficient between training accuracy and child network final accuracy is 0.79. We didn't run and evaluate DARTS for statistically sufficient number of times to reach any claim of the correlation. Could you please provide some justification for that? As if this claim was true, it would help validate our claim that the manually designed child network derivation scheme is biased...
>
> 5) Report results from more runs
> We believe the reported result can support our claims. Nonetheless, we are evaluating more child networks to provide this variance. Thank you for your suggestion!

---

### Author Response · Authors · 2018-11-20
**Manuscript updated**

We thank all reviewers for your recommendation, comments and expressing your concerns. We have updated the manuscript in Section 1, Section 3.1 and Section 3.2, taking your feedbacks into account. Here we provide a summary of these updates:

1) We have tried extensive sweeps on the constraint hyperparameter \eta. Previously we only tried \eta that lies at the margin of appearance of ZERO op in the child network. The new discovery is that with a larger \eta, the regularizing effect of resource constraint becomes obvious. A pair of new cells was discovered on CIFAR-10, which achieves better accuracy than 1st-order DARTS, as well as ENAS. Its accuracy is also on par with 2nd-order DARTS, with fewer parameters. In the updated version, we report its accuracy and parameter size, with the architecture attached in Appendix H.

2) When a more aggressive constraint is applied, more edges are dropped in SNAS. A new figure is added to exhibit SNAS's capability of discovering sparse structures that ENAS and DARTS are not able to discover. With 1/3 fewer parameters, it achieves on-par accuracy with 1st-order DARTS.

3) We updated figure 4 as ZERO was not excluded in the previous version. As discovered by some reviewer, it is omitted during operation selection for child graph in the code released by DARTS's authors. Besides, since to use variance to measure how much architecture weights differentiate from each other might be confusing, we updated it with entropy of softmax at every edge, which we hope to be more self-explanatory. Basically, the conclusion remains the same, in SNAS the learnt architecture distribution is more certain about the structural decisions.

---

> ### Author Response · Authors · 2018-11-26
> **Manuscript updated**
>
> We have updated Table 2 to include results from three runs of SNAS as requested by one reviewer.

---

> > ### Author Response · Authors · 2018-11-26
> > **Manuscript updated**
> >
> > We have updated Table 2 to include results of DARTS with single-level optimization as reported by DARTS’s authors for fair comparison with SNAS. SNAS is single-level optimization because it simultaneously optimizes neural operation parameters and architecture distribution parameters over the same dataset. Analysis is included in Section 3.3 Results. For further details, please refer to the response to AnonReviewer1 [1].
> >
> > [1] https://openreview.net/forum?id=rylqooRqK7&noteId=HJgkGl3FRQ

---

### Public Comment · (anonymous) · 2018-11-30
**Code Release?**

Will you be releasing code for SNAS?  It would help fill in some of the details for how the architecture distribution parameters are trained.

---

> ### Author Response · Authors · 2020-04-01
> **Code has been released.**
>
> Thanks for your interest, we have released our implementation at https://github.com/SNAS-Series/SNAS-Series.

---

### Meta-Review · Area_Chair1 · 2018-12-13
**Alternative way to differentiable NAS**

**Confidence:** 4
**Recommendation:** Accept (Poster)

**Metareview:**

This paper provides an alternative way to enable differentiable optimization to the neural architecture search problem.  Different from DARTS, SNAS reformulates the problem and employs Gumbel random variables to directly optimize the NAS objective. In addition, the resource-constrained regularization is interesting. The major cons of the paper is that the empirical results are not quite impressive, especially when compared to DARTS, in terms of both accuracy and convergence. I think this is a borderline paper but maybe good enough for acceptance.